# Nanomaterial-Mediated Theranostics for Vascular Diseases

**Swati Agrawal [1], Sunil K. Nooti [2], Harbinder Singh [2] and Vikrant Rai [2,***

[1]  Department of Surgery, Creighton University School of Medicine, Omaha, NE 68178, USA; swatiagrawal@creighton.edu
[2]  Department of Translational Research, Western University of Health Sciences, Pomona, CA 91766, USA; snooti@westernu.edu (S.K.N.); singhh@westernu.edu (H.S.)
*   Correspondence: vrai@westernu.edu

**Abstract:** Nanotechnology could offer a new complementary strategy for the treatment of vascular diseases including coronary, carotid, or peripheral arterial disease due to narrowing or blockage of the artery caused by atherosclerosis. These arterial diseases manifest correspondingly as angina and myocardial infarction, stroke, and intermittent claudication of leg muscles during exercise. The pathogenesis of atherosclerosis involves biological events at the cellular and molecular level, thus targeting these using nanomaterials precisely and effectively could result in a better outcome. Nanotechnology can mitigate the pathological events by enhancing the therapeutic efficacy of the therapeutic agent by delivering it at the point of a lesion in a controlled and efficacious manner. Further, combining therapeutics with imaging will enhance the theranostic ability in atherosclerosis. Additionally, nanoparticles can provide a range of delivery systems for genes, proteins, cells, and drugs, which individually or in combination can address various problems within the arteries. Imaging studies combined with nanoparticles helps in evaluating the disease progression as well as the response to the treatment because imaging and diagnostic agents can be delivered precisely to the targeted destinations via nanocarriers. This review focuses on the use of nanotechnology in theranostics of coronary artery and peripheral arterial disease.

**Keywords:** coronary artery disease; peripheral arterial disease; inflammation; atherosclerosis; nanoparticles; nanocarriers; nanomaterials; nanotechnology; theranostics

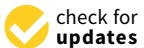



## 1. Introduction

Nanotechnology comprises the study and design of minute materials and machines. The term "Nano" originated from the Greek word meaning dwarf [1]. The tunable shape, size, and surface chemistry of the nanoparticles makes them a powerful tool for drug delivery and defined distribution with selectively targeting the lesion or the mediators involved in the pathogenesis. This increases the efficacy of the drug and decreases the side effects and off-target effects. Commonly used nanomaterials for drug delivery are polymeric nanoparticles, polymer micellar co-delivery systems, solid lipid nanoparticles, liposomes, metallic nanoparticles, inorganic non-metallic nanomaterials, and dendrimers [2,3]. Nanoparticle-based therapy has long been used in cancer treatment and currently, more than 50 nanoparticle-based therapies are used for the treatment of infection, chronic inflammation, chronic kidney diseases, psychiatric conditions, developmental, and degenerative nervous system disorders [2,3]. The use of nanotechnology in treating cardiovascular disorders has also been expanded, for example, the use of nano-formulations of fenofibrate to overcome challenges with drug solubility and absorption in treating hypertriglyceridemia [3]. The nano-drug carriers have their advantages and disadvantages. Good biocompatibility, good stability, non-immunogenicity, ease of preparation, antibacterial properties, stability, large surface area, and pore volume are advantages, while toxicity, leakage of hydrophilic drugs, and depolymerization after dilution are some of the disadvantages [4]. This review focuses on the role of nanotechnology in the treatment of vascular diseases including coronary artery disease and peripheral arterial disease.

## 2. Vascular Disease and Nanomedicine

Coronary artery disease (CAD) is characterized by narrowing of the artery due to atherosclerosis and leads to clinical symptoms including angina and myocardial infarction. Similarly, atherosclerosis of the carotid artery leads to cerebral ischemia and stroke. Atherosclerosis develops from a fatty streak progressively developing to atheroma, atheromatous plaque, fibroatheroma, and finally to stable plaque (Figure 1). The development of atherosclerosis is associated with risk factors including hypercholesterolemia, hyperglycemia, hypertension, smoking, male sex, and a family history. Inflammation in the stable plaque results in unstable plaque characterized by a thin fibrous cap, necrotic core, and proliferation of vascular smooth muscle cells, angiogenesis, and calcification, which is prone to rupture, leading to ischemic events [5,6]. Mediators of inflammation, including high mobility group box 1 (HMGB-1), receptor for advanced glycation end products (RAGE), toll-like receptors (TLRs), triggering receptor expressed on myeloid cells-1 (TREM-1), proinflammatory cytokines including interleukin (IL)-1$\beta$, IL-6, tumor necrosis factor (TNF)-$\alpha$, and immune cells such as macrophages, have been found to be involved in the pathogenesis of atherosclerosis [5,7–10]. Vascular diseases including coronary artery disease and peripheral vascular diseases involve vascular inflammation and development of localized atherosclerotic plaque as a lesion; targeting localized inflammation and involved specific inflammatory mediators might play a crucial role in attenuating the progression of plaque and treatment of the disease. Reducing low-density lipoprotein (LDL) with statins, omega-3 fatty acids, niacin, and fibrates, targeting mediators of inflammation, and vitamin D supplementation have been discussed as treatment strategies. Angioplasty is the available surgical treatment but is associated with restenosis as a complication and research is on-going to either attenuate the formation of atherosclerotic plaque or prevent restenosis after angioplasty [7,11–13]. Statins have shown a reduction in morbidity and mortality in atherosclerotic disease [14]; however, no definitive pharmacological treatment has been postulated. Issues with water solubility, poor bioavailability, low biological efficacy, non-targeting, off-target effects, systemic exposure, drug resistance, and drug–drug interactions remain a concern in treating cardiovascular diseases with statins, antithrombotic, and thrombolytic agents. Moreover, achieving a site-specific and controlled release of therapeutics at a specific inflammatory site remains a big challenge. Since atherosclerosis involves pathogenesis at the molecular and cellular levels, effective treatment at the cellular level might be beneficial, and delivering drugs at the cellular level with nanotechnology makes nanomedicine an effective complementary therapeutic. Using nanomedicine to treat the localized and target- and site-specific inflammation more efficiently, with an anatomical peculiarity to deliver the drug, might be a potential strategy [15,16].

Peripheral vascular disease is an umbrella term consisting of peripheral arterial disease affecting arteries and diseases of the peripheral veins. Peripheral arterial disease (PAD) is also a manifestation of a decreased arterial lumen distal to the arch of the aorta due to atherosclerosis and decreased blood supply leading to hypoxia and ischemia in the peripheries. Intermittent claudication presenting as pain in the muscles of the legs with exercise is the most common clinical symptom of PAD. Common signs include abnormal pedal pulses, delayed venous filling time, femoral artery bruit, cool skin, and abnormal skin color [17]. Reducing platelet aggregation, managing hypertension, diabetes, and hyperlipidemia, using stents, angioplasty, arterectomies, and bypass surgery are commonly available treatments. Using growth factor (genes or proteins) and cell-based therapy to the ischemic tissue to stimulate the regeneration of functional vasculature and re-perfusing the ischemic tissue is an emerging alternative strategy to save the limb. These therapies consist of cell therapy using bone marrow mononuclear cells (BMMNCs), mesenchymal stem cells (MSCs), induced pluripotent stem cell (iPS), G-CSF-mobilized peripheral blood mononuclear cells (M-PBMNCs), janus magnetic cellular spheroid (JMCS), and endothelial progenitor cells (EPCs) and using proangiogenic factors, such as vascular endothelial growth factor (VEGF), fibroblast growth factor (FGF), and hepatocyte growth factors (HGF) [18–20]. Gene therapy and cell-based therapies can be facilitated and improved

using nanomaterials. Nanoparticles can be a fascinating drug carrier with efficient delivery and can target the ischemic tissues. This strategy will also aid in achieving localized and sustained release of pro-angiogenic genes and proteins [21]. Nanoparticles combined with cell therapy will help to enhance cell retention, cell survival, and secretion of angiogenic factors. Further, the use of nanotechnology-based stents or grafts (drug-eluting stents, synthetic stents, and autologous grafts) will overcome many issues due to their better structural integrity mimicking the natural vessel.

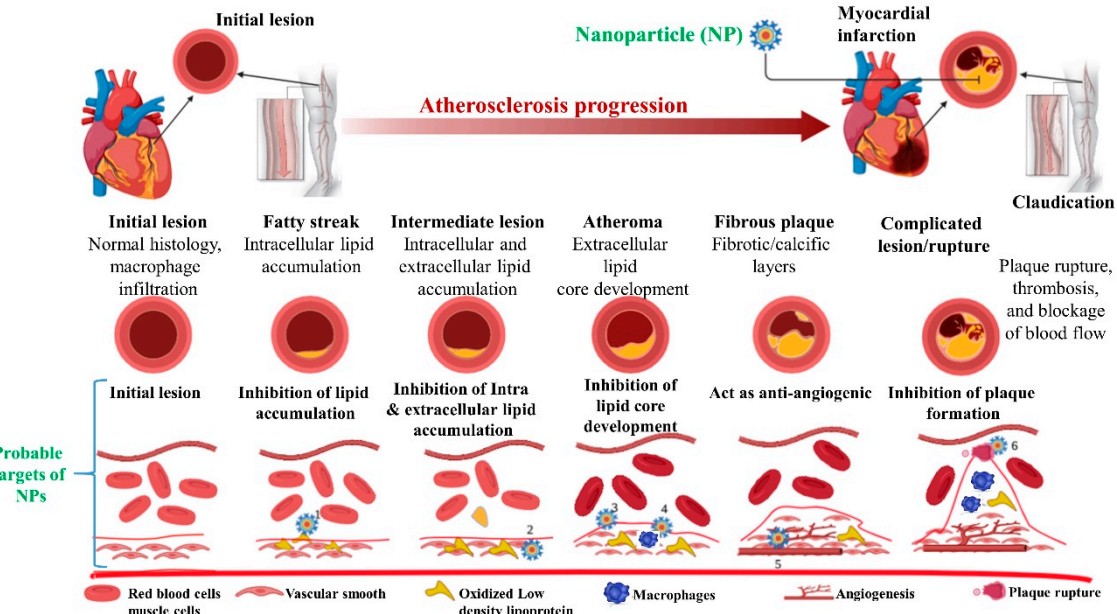

**Figure 1.** Development of atherosclerosis and targets of therapeutic importance using nanoparticles: The pathogenesis of atherosclerosis starts with fatty streak with increased deposition and oxidation of lipids and infiltration of immune cells results in the formation of atheroma, fibroatheroma, and atheromatous plaque (stable) followed by the unstable plaque that is prone to rupture. Nanomaterials might be used to target various factors involved in the pathogenesis of atherosclerosis including lipid metabolism (1), vascular smooth muscles (2), endothelial cells (3), macrophages (4), angiogenesis (5), and thrombus (6) to deliver the growth factors, genes, proteins, and drugs. Neointimal tissue and extracellular membrane components are other potential targets as discussed in the text.

## 3. Therapeutic Role in Vascular Diseases

The treatment of CAD using nanotechnology may involve non-invasive techniques aimed to reduce lipid levels, targeting angiogenesis, inflammation, intra-arterial thrombus, and invasive techniques such as anti-restenosis treatment after the percutaneous coronary intervention (PCI), prevention of in-stent restenosis, enhancement of healing after PCI, and using biosynthetic CABG grafts [22]. Targeting mediators of inflammation and inflammatory pathways to attenuate inflammation have been described as a therapeutic strategy to attenuate the progression of atherosclerosis and prevent rupture of unstable plaque. Nanotechnology can be used to target these pathways and various delivery methods are under development targeting thrombosis, neointima formation, lipid metabolism, macrophages, and plaque formation (Figure 1). These include IL-10 nanoparticles targeting collagen IV, tissue plasminogen activator (tPA) nanoparticles to target von Willebrand factor (vWF) in thrombi, nanoparticle stabilized delivery of proprotein convertase subtilisin/kexin type 9 (PCSK9) siRNA in hepatocytes, paclitaxel nanoparticle targeting injured endothelium to prevent the growth of neointima, antiangiogenic nanoparticles targeting $\alpha_v\beta_3$ on neointima to prevent neovascularization, and nanoparticles targeting CD40 (cluster of differentiation 40)-induced TRAF6 (tumor necrosis factor receptor-associated factor 6) signaling in macrophages, and others as discussed in [3]. Inhalation, oral administration, or intravenous injection are common routes for nanomaterial mediated drug administration.

Inflammation is involved in the pathogenesis of atherosclerosis of arterial diseases including coronary artery disease, carotid atherosclerosis, and peripheral arterial disease. The inflamed site of the vascular wall is under oxidative stress and acidosis due to reactive oxygen species (ROS) overproduction from endothelial cells (ECs) and vascular smooth muscle cells (VSMCs) through the activity of several enzymes including xanthine oxidase, mitochondrial respiratory chain enzymes, and a dysfunctional, uncoupled endothelial nitric oxide synthase (eNOS), but mainly nicotinamide adenine dinucleotide phosphate (NADPH) oxidases [23]. Targeting oxidative stress and ROS to enhance cell survival has been documented [24]. Based on these facts, Zhang et al. aimed to target acidosis (pH) and oxidative stress (ROS) simultaneously with active targeting nanoparticles and reported significant efficacy and potency of collagen IV targeting, pH/ROS dual-responsive RAP/TAOCD nanoparticles (made of rapamycin-RAP, pH-sensitive material-ACD, and oxidation-responsive material-OCD) in in-vitro and in-vivo (mice and rat injected intravenously) leading to inhibition of neointima formation. The study suggested that these nanoparticles are an effective and safe nano-vehicle for precise and targeted therapy in vascular inflammation and arterial restenosis [16].

Oxidized LDL and inflammation are involved in the pathology of atherosclerosis; targeting macrophages, oxidized low-density lipoprotein (oxLDL), and microvessels might be useful in attenuating atherosclerosis. Recently, magnetic resonance imaging (MRI) with iron oxide nanoparticles has been extensively probed as a tool to detect inflammation in high-risk plaques vulnerable to rupture [25]. Iron oxide nanoparticles contain an iron oxide core and based on the size, they can be used for intravascular versus tissue imaging. The local magnetic field inhomogeneities due to tissue accumulation of these nanoparticles are measured by MRI. These nanoparticles can be labeled with radioisotopes or fluorophores to increase their diagnostic properties. The detection of iron oxide nanoparticles phagocytized by inflammatory cells and reticuloendothelial system cells may indicate the inflammatory cells involved in the process of inflammation and atherosclerosis [25]. Ferumoxtran-10 are ultra-small superparamagnetic iron oxides (SPIO) approved for treating iron-deficient anemia in chronic kidney disease and their uptake has been demonstrated in the symptomatic and contralateral asymptomatic carotid artery [26] and coronary artery disease [27]. A decreased uptake of ferumoxtran-10 in human carotid arteries after 12 weeks of treatment with high dose atorvastatin in the ATHEROMA study (Atorvastatin Therapy: Effects on Reduction of Macrophage Activity) is indicative of attenuated inflammation and the utility of ferumoxtran-10 in assessing the treatment response [28] but the lack of correlation between ferumoxtran-10 uptake and carotid stenosis reported by Tang et al. [29] suggests its diagnostic paucity. Carotid restenosis after coronary artery angioplasty is due to mechanical injury, inflammatory response, and delayed endothelial healing, and thus, preventing restenosis is the focus of current research. The use of nanoparticle-based drug-eluting stents to prevent restenosis has been reported. Nakano et al. [30] reported the use of bioabsorbable polymeric nanoparticle-eluting stents for efficient nano-drug delivery in porcine coronary arteries and suggested using NP-eluting stents as an efficient and innovative NP-mediated drug delivery system. Tsukie et al. [31], in a porcine coronary artery model, reported the use of pitavastatin-incorporated nanoparticle-eluting stents to attenuate in-stent stenosis. The study found that pitavastatin-NP-eluting stents attenuated in-stent stenosis as effectively as polymer-coated sirolimus-eluting stents (SES) but delayed endothelial healing effects were noted with SES and not with pitavastatin-NP-eluting stents. These results suggest an important role of nanoparticles in the treatment of atherosclerosis and inhibition of restenosis (Table 1). Similarly, the role of paclitaxel-loaded magnetic nanoparticles (PTX-MNPs), VEGF-loaded, IR800-conjugated, graphene oxide (GO-IR800-VEGF) nanoparticles, rhodamine B- labeled, PEGylated R-SiNPs for sustained delivery of VEGF in a murine model, bivalirudin-functionalized perfluorocarbon nanoparticles (PFC-NPs), and EDTA- and PGG-loaded nanoparticles [32] has been reviewed.

**Table 1.** Nanoparticles used for therapy and diagnosis in carotid and aortic arterial disease. High-density lipoproteins (HDL), nanoparticles (NPs), nitric oxide synthase cofactor tetrahydrobiopterin (BH4), p38 mitogen-activated protein kinases (MAPK), scavenger receptor-AI (SR-AI), ultrasmall superparamagnetic iron oxide (USPIO), and paclitaxel and functionalized the particles with collagen targeting specific C11 polypeptide (UP-NP-C11), extra domain B of fibronectin (FN-EDB).

| Study | Nanoparticles | Animal Model | Outcome |
|---|---|---|---|
| [33] | P904 | Hereditary hyperlipidemic rabbit | P904 accumulation and low endothelial shear stress are independent predictors of plaque progression |
| [34] | Dextran-coated monocrystalline iron oxide | Hereditary hyperlipidemic rabbit | Accumulation in vessel wall correlating with plaque macrophages and the accumulation reduced with rosuvastatin |
| [35] | CLIO-CyAm7 | Rabbit with atherosclerosis induced by aortic balloon injury and high-cholesterol diet | Significantly higher CLIO-CyAm7 accumulation in thrombosed than in nonthrombosed plaques |
| [36] | USPIO | ApoE$^{-/-}$ mice model of atherosclerosis | Noninvasive assessment of USPIO uptake is a marker for inflammation in murine atherosclerotic plaque Reduction in ferumoxtran-10 uptake after treatment with p38-MAPK inhibitor |
| [37] | USPIO conjugated to SR-AI ligand | ApoE$^{-/-}$ mice model of atherosclerosis | SR-AI–targeted USPIO displayed accelerated plasma decay and a 3.5-fold increase accumulation in atherosclerotic plaque SR-AI–targeted molecular imaging of USPIO-based contrast may be used to detect inflammatory plaques |
| [38] | Feridex-a dextran-based USPIO | High cholesterol fed rabbits | 15 nm fractionated, but not nonfractionated feridex gets deposited in atherosclerotic plaques and thus size should be further investigated |
| [39] | HDL NPs collagen-specific EP3533 peptides (EP3533-HDL) | Reversa mouse model of atherosclerosis regression | HDL NPs may be used to monitor and evaluate compositional changes in atherosclerotic plaque regression |
| [40] | FN-EDB-specific Gd NPs (APT$_{FN-EDB}$-[Gd]NP) | Murine ApoE$^{-/-}$ model of atherosclerosis | Augmented FN-EDB expression in Type III, IV, and V atheroma and these NPs may be used to identify and/or deliver agents locally to a subset of atherosclerotic plaques. |
| [41] | Cathepsin-B activatable L-SR15 | Apolipoprotein E knock-out atheromatous mouse model | Selective apoptotic attenuation of macrophages Reduction in cathepsin-B protease activity |
| [42] | Gold nanorods (Au NRs) | Apolipoprotein E knockout mice model with femoral artery restenosis | Ablation of inflammatory macrophage layer in Au NRs group compared to the controls Au NRs are effective for in-vivo imaging and photothermal therapy of inflammatory macrophages |
| [43] | UP-NP-C11 | Rabbit | Significantly higher accumulation of NPs atherosclerotic plaques compared to the control condition |

**Table 1.** *Cont.*

| Study | Nanoparticles | Animal Model | Outcome |
|---|---|---|---|
| [44] | Statin-rHDL NPs | ApoE-knockout mouse model of atherosclerosis | 3-month low-dose statin-rHDL treatment inhibits plaque inflammation progression 1-week of high-dose regimen markedly decreases inflammation in advanced atherosclerotic plaques |
| [45] | Paramagnetic NPs loaded with anti-angiogenic drug fumagillin | Cholesterol-fed rabbit | Targets $\alpha v \beta 3$ integrin Showed an anti-angiogenic response compared to controls Molecular imaging combined with drug delivery with NPs noninvasively define atherosclerotic burden and response to treatment |
| [46] | PREY-nanocarriers loaded with BH4 | Fat-fed atheroprone mice $(ApoE^{-/-})$ | Reduced plaque burden in partially ligated left carotid artery A potential strategy to prevent atherosclerotic plaque formation |

Increased accumulation of nanoparticles in the ischemic tissue compared to the non-ischemic limb and the use of nanomaterials including poly(lactic-co-glycolic acid) (PLGA), polycaprolactone (PCL), poly(lactic) acid (PLA), poly(sebacic) acid-polyethylene glycol (PSA-PEG), gold nanoparticles and graphene oxide to deliver growth factors to ischemic tissues have been discussed [20,21,47]. Combining the use of growth factors or cell-based therapy may enhance the efficacy and potency and help to design a specific release kinetic profile. Significantly increased targeting efficiency of VEGF-coated graphene oxide nanoparticles compared to empty nanoparticles supports the notion that VEGF coating serves both as a therapeutic reagent and as a targeting moiety [47]. Varying growth factor release dynamics have been reported in in-vitro studies such as a near zero-order release of VEGF with dextran-co-gelatin nanoparticles with 69% VEGF released over 10 days [48], 50% released in the first 8 days, and steady release of bFGF up to 20 days with mesoporous silica nanoparticles [49], and 70% of encapsulated VEGF released within 2 days with PLGA nanoparticles [50]. The role of nanoparticles using growth factors in animal models to increase vascularity using dextran-co-gelatin nanoparticles encapsulating VEGF, VEGF conjugated on gold nanoparticles, delivering VEGF with graphene oxide nanoparticles, and liposomal codelivery of FGF-2 with syndecan-4 has been discussed in the literature [21]. Nanoparticles such as magnetic DNA-gelatin nanospheres, PLGA nanoparticles, PEG liposomes, PEG liposomes containing DOTAP, peptides-DNA nanoparticles, and heparinized chitosan/poly($\gamma$-glutamic acid) nanoparticles have been used to deliver plasmid DNA of VEGF, bFGF, and HIF-1$\alpha$ in rabbit and mouse models and have shown promising results of increased blood flow [21]. The use of liposomes for site-selective delivery [51], chitosan in decreasing tissue necrosis [52], PLGA in attenuating VSMC proliferation, inhibiting restenosis, and increasing blood flow [53], PVAX for increased cell viability, cell migration, and blood flow [54], and HPOX to increase blood flow and augmented endothelial cell recruitment and proangiogenic growth factor secretion [55] suggest the role of nanomaterials in therapeutics of vascular diseases. Further, atherosclerosis is a multifactorial disease and targeting multiple targets might be a potential therapeutic strategy and nanomaterial-mediated multi-drug delivery might be beneficial for the treatment of vascular disease [24].

## 4. Diagnostic Role in Vascular Diseases

Nanotechnology can be used not only in therapeutics but also in diagnostics, which means it holds potential in both the delivery of drugs and imaging agents. Using gold

nanoparticles in the form of contrast agents in photoacoustic imaging and optical imaging of coronary blood vessels, conjugated radiotracers with nanoparticles in imaging (using the advantage of increased signal to noise ratio), and ferromagnetic iron oxide particles with poly-dispersive properties to accentuate the contrast for MRI enhances the diagnostic accuracy (Figure 2).

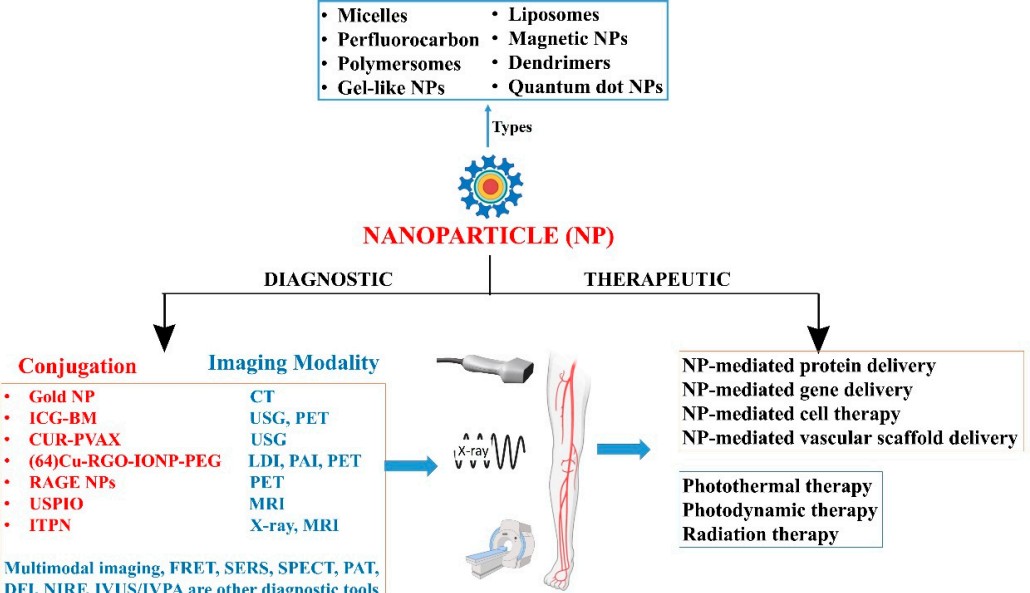

**Figure 2.** Nanoparticles in diagnosis of peripheral arterial diseases: computerized tomography (CT), single-photon emission computerized tomography (SPECT), positron emission tomography (PET), magnetic resonance imaging (MRI), ultrasonogram (USG), Förster resonance energy transfer (FRET), surface-enhanced Raman spectroscopy (SERS), near-infrared fluorescence (NIRF) imaging, photoacoustic tomography (PAT), dark field imaging (DFI), photoacoustic imaging (PAI), laser device imaging (LDI).

Additionally, use of nanoparticle-based blood pool contrast agents to visualize vasculature in vivo, polydispersive iron particles for drug delivery at lower magnetic field gradients, quantum dots in membrane protein labeling and cell tracking studies, fluorescence resonance energy transfer (FRET) in assessing enzymatic activity in atherosclerosis suggests the role of magnetic nanoparticles in imaging, sensing, tagging, and separation [22]. Nanomaterial can also be used to distinguish between stable and unstable plaque prone to rupture [56]. The theranostic nanoparticles (combined therapeutic and diagnostic nanoparticles) may be used to combine even multiple drugs targeting inflammation and imaging modalities [3]. The lowered accumulation and fast-food clearance of P904, an ultra-small superparamagnetic iron oxide nanoparticle, in aortic plaques of atherosclerotic mice with irbesartan [57] and accumulation of ferumoxtran-10 and ferumoxytol in rabbit aortic plaques [58] suggest the diagnostic role of nanoparticles. Imaging the structural and cellular components of atherosclerotic plaques with greater definition and improved sensitivity will help in planning better therapeutic strategies by tailoring pharmacokinetic parameters that allow for enhanced pharmacodynamic effects. This will also allow for monitoring the disease progression and progress of the treatment (Table 1).

Even though there have been numerous studies focusing on the diagnostic use of nanoparticles in coronary artery disease, there are a few notable articles which have focused on the use of nanoparticles in peripheral arterial disease (PAD). In Table 2, we summarize the various targeted nanoparticles developed for the purpose of diagnostic evaluation and or therapy of peripheral arterial disease in animals and human patients.

**Table 2.** Nanoparticles used for diagnosis and therapy in peripheral arterial disease.

| Nanoparticle | Mechanism | Imaging Method | Outcome |
|---|---|---|---|
| Indocyanine green-loaded boronated maltodextrin (ICG-BM) nanoparticles [59] | Real time detection of ROS through $H_2O_2$-activatable $CO_2$ bubble generation by ICG-BM | Photoacoustic imaging with an ultrasound machine and a pulsed laser system | ICG-BM nanoparticles could be used as multiple contrast agents for enhanced fluorescence, ultrasound, and photoacoustic imaging. |
| Curcumin (CUR) loaded vanillyl alcohol-incorporated copolyoxalate (PVAX) nanoparticles (CUR-PVAX) [60] | Generation of echogenic $CO_2$ bubbles through $H_2O_2$ | Ultrasound | CUR-PVAX enhances the ultrasound signal and also act therapeutically to suppress expression of pro-inflammatory cytokines. |
| (64)Cu-labeled PEGylated reduced graphene oxide—iron oxide nanoparticles ((64)Cu-RGO-IONP-PEG) [61] | Passive accumulation in ischemic tissues through enhanced permeability and retention (EPR) effect. | Laser Doppler imaging, Photoacoustic imaging, and PET | Quantitative confirmation of accumulation of PEGylated nanoparticles in Ischemic tissues of hindlimb. |
| PEGylated long circulating organic-inorganic hybrid nanoparticles [62] | Passive accumulation in ischemic tissues with reduced ABC (accelerated blood clearance) effect. | Positron emission tomography (PET) | Validation of ABC phenomenon |
| Receptor for advanced glycation end products (RAGE) multimodal nanoparticle [63] | Specific molecular targeting of RAGE expressed in hindlimb ischemia murine model | PET | Non-invasive examination of cellular, tissue and whole body RAGE levels is feasible |
| biodegradable poly(β-amino ester) (PBAE)-based CXCR4 pDNA nanoparticles [64] | Polyester nanoparticles enhance transfection efficiency of adipose-derived stem cells (ADSCs) in vivo. | Bioluminescence imaging of the GFP labeled cells and laser Doppler system for blood flow. | Complete limb salvage in a mouse ischemia limb model. |
| ultrasmall superparamagnetic iron-oxide (USPIO) (ferumoxytol) nanoparticles [65] | Long circulating nanoparticles such as USPIOs are taken up by tissue macrophages that can be imaged in plaques using MRI | Dynamic contrast-enhanced (DCE)-MRI using gadolinium-based contrast agents | Confirmation of accumulation of USPIOs in atherosclerotic plaques, assessed by quantitative DCE-MRI in PAD patients. |
| Nitric Oxide releasing nanoparticles (NO-nps) [66] | Delivery of NO with nanoparticles | NA | Stabilization of hemodynamics preservation of micro vascular perfusion in acute hemorrhage |
| alpha(nu)beta(3)-integrin-targeted perfluorocarbon nanoparticles. [67] | Alpha(nu)beta(3)-integrin is a biomarker for neovascular proliferation in angiogenesis. | MRI and X-ray angiography | In vivo (rabbit femoral artery) demonstration of non-invasive molecular imaging of angiogenesis. |

Conventionally, PAD detection is performed by ankle brachial index or ABI using a Doppler ultrasound device (uncommonly Toe Brachial Index or TBI and angiography are also used for PAD diagnosis). However, these have many limitations including the accuracy

of the severity of disease especially in calcified vessels, patients with poor blood flow and obesity. Detection of PAD with biomarkers such as serum bilirubin [68], *N*-terminal pro-B type natriuretic peptide [69], plasma C-reactive protein, von Willebrand factor (vWF), interleukin 6, red cell folate (RCF), vitamin B12, and total homocysteine (tHcy), as reviewed in [70], is still in its infancy. However, NPs are versatile for in vivo imaging, because they carry a larger amount of contrast agents than a single small molecule while reducing systemic toxicity. Nanoparticles are also credited for their easy surface modification and high targeting capacity and as such have been used for PAD. Iron oxide NPs due to their elevated magnetization properties are used as T2 weighted agents with MRI. Monocrystalline iron oxide nanoparticles (MION-47), superparamagnetic iron oxide (SPIO), ultra-small superparamagnetic iron oxide (USPIO) and lipid-coated ultra-small superparamagnetic iron particles (LUSPIOs) have all been used for quantification of accumulated foam cells in plaques and their progression in vivo [34,71,72]. These are usually conjugated with specific targets amenable to accumulation at the lesioned arteries and usually are pro-inflammatory molecules, which are elevated in the diseased arteries. Ischemic tissues generate reactive oxygen species (ROS) of which hydrogen peroxide ($H_2O_2$) is a major component that can be used by $H_2O_2$ activatable $CO_2$ bubble generating indocyanine green-loaded boronated maltodextrin (ICG-BM) nanoparticles [59] and curcumin (CUR)-loaded vanillyl alcohol-incorporated copolyoxalate (PVAX) nanoparticles (CUR-PVAX). CUR-PVAX nanoparticles also act to decrease the expression of pro-inflammatory cytokines [60]. The receptor for advanced glycation end-products (RAGE) is a marker for chronic inflammation such as that occurring in PAD with diabetes. A multimodal nanoparticle based imaging agent specifically targeting RAGE was utilized in ischemic RAGE-expressing hind limbs, where the uptake was significantly enhanced compared to their non-ischemic counterparts [63]. Nanoparticles can also passively accumulate in ischemic tissues through the enhanced permeability and retention (EPR) effect, which was utilized by England et al. [61] with (64)Cu-labeled PEGylated reduced graphene oxide—iron oxide nanoparticles ((64)Cu-RGO-IONP-PEG) to quantify the same in a murine model of PAD. While utilizing pegylated nanoparticles it was shown that multiple injections into the same individual shortened the circulation half-lives in vivo, a phenomenon known as accelerated blood clearance (ABC), which is a concern for clinical translation [62]. Nanoparticles are also efficient enhancers of plasmid DNA transfection in stem cells for in vivo transplantation and this attribute has been used in overexpressing CXCR4 in adipose derived stem cells to completely salvage an ischemic limb in a murine model [64].

The role of nanoparticles such as gold nanoparticles, EP3533-HDL-like nanoparticles, Gd-CREKA-targeted peptide amphiphilic micelles, CREKA-based iron oxide nanoparticles, CLIO-CyAM7, iron oxide nanoparticles conjugated to polyclonal antibodies against profilin1 (PC-NPs), USPIO nanoparticles targeted to lectin-like oxidized LDL receptor 1, $^{19}F$ perfluorocarbons nanoparticles, etc. in diagnosis and therapeutics has been discussed [73,74] (Table 1). Additionally, the role of ultra-small paramagnetic iron oxide (USPIO), lipid ultra-small paramagnetic iron oxide (LUSPIO), iodinated nanoparticles, paramagnetic perfluorocarbon nanoparticles (PM-PFCNPs), superparamagnetic iron nanoparticles (MION-47), superparamagnetic iron oxide (SPIO), liposomes with magnetic nanoparticle (MNP), and others in detecting PAD has been reviewed [20] (Table 1). 5-Aminolevulinic acid-derived PPIX accumulates in atherosclerotic plaques and its fluorescence intensity positively correlates with plaque macrophage content [75]. Based on this, Gonçalves et al., [76], using a rabbit animal model, evaluated the theranostic role of PEGylated ALA gold nanoparticles (ALA:AuNPs) in clinical practice for atherosclerosis and based on the findings of the rapid conversion of ALA into endogenous porphyrins, the study suggested that this strategy can help in the early diagnosis and treatment of atherosclerosis. The enhanced detection power of nanoparticles is due to enhanced binding affinity, their specificity, amplified signals, and increased imaging capacity by combining several imaging labels for multiple imaging modalities with a single nanoparticle [77]. Additionally, the advantage of combining these nanoparticles with imaging such as MRI, intravascular ultrasound and

photoacoustic (IVUS/IVPA), positron emission tomography (PET), CT scan, optical coherence tomography (OCT), nuclear imaging, optical imaging, single-photon emission computed tomography and contrast agents to promote the diagnostic values have also been discussed in the literature [74].

## 5. Advantages and Limitations

Delivering drugs with nanotechnology might be advantageous to wean off the issues of systemic exposure, drug–drug interactions, off-target effects, poor bioavailability, water-solubility, low biological efficacy, and drug resistance. Nanomaterial-mediated drug delivery improves safety and effectiveness by increasing drug stability and water solubility, increasing the drug uptake by target cells, prolonging the cycle time, and reducing enzymatic degradation [78,79]. Nanoparticle-based therapy also helps in overcoming biological barriers and enhancing the therapeutic index of the drug [3]. Additionally, increased vascular permeability and decreased lymphatic drainage mediated by inflammation facilitate increased retention and duration of the effect of nanomedicines [80]. There were exciting results from the use of silica-gold nanoparticles for atheroprotective management of plaques in a first-in-man trial (the NANOM FIM trial NCT01270139) with three patient groups (1) nano-intervention with delivery of silica-gold NP in a bioengineered on-artery patch ($n$ = 60 nano group), or (2) nano-intervention with delivery of silica-gold iron-bearing NP with targeted micro-bubbles and stem cells using a magnetic navigation system ($n$ = 60 ferro group) versus (3) stent implantation ($n$ = 60 stenting group), which showed a significantly lower risk of cardiovascular death in the first group compared with others (91.7% vs. 81.7% and 80% respectively; $p < 0.05$) [81]. With respect to long-term outcomes, a prospective cohort analysis of the 5-year clinical outcomes at the intention-to-treat population (nano vs. ferro vs. stenting; $n$ = 180) led to a better rate of mortality, major adverse cardiovascular events and target lesion revascularization [82] in the nano group.

Nanomedicine seems to be an effective therapy; however, there are challenges in implementing it. Aseptic conditions or even the tiniest contamination during the development of nanomachines, nanotubes, nanowires, and nanospheres may lead to adverse effects in patients [22]. The toxicity of the nanomaterials due to its chemical and mechanical structure, dimensions, surface coating, and DNA damage are other concerns while using nanomaterials [83–85]. Further, the age of nanomaterials in the biological system is not determined and no definitive answer is there about the potential side effects of the nanomaterial and nanomachines at the cellular level [86,87]. This complexity is due to complex synthesis, long-term toxicity of nanomaterials, the low sensitivity of MRI agents, and the absence of a human disease model. Thus, no nanomaterial is commercially available to use in clinics for treatment of atherosclerosis and not a single nanomaterial has satisfied the criteria of inexpensive synthesis, low toxicity, and ease of manufacturing [88]. Based on the above studies, the advantages and limitations are listed in Table 3. Additionally, the advantages and limitations and disadvantages of current techniques for analyzing biodistribution of nanoparticles and diagnostic modalities including histology, electron microscopy, liquid scintillation counting, measuring drug concentration in tissue, computed tomography, MRA, and PET scans have been reviewed elsewhere [89]. Thus, there is a need to collaboratively and critically work on improving the nanomaterials and the strategies to target the pathology in PAD and implementing and translating these technologies for a positive clinical outcome.

**Table 3.** Advantages and disadvantages of nanotechnology.

| Advantages | Disadvantages/Limitations |
|---|---|
| ➢ Site-specific targeted delivery | ➢ Possibilities of contamination |
| ➢ Faster and accurate delivery | ➢ Nanoparticle-mediated infection/sepsis |
| ➢ Feasibility of therapy at cellular level | ➢ Nanomaterial-related toxicity |
| ➢ Refined drug production | ➢ Issues with biodegradability of nanomaterials |
| ➢ Drug tailoring at molecular level | ➢ Complex synthesis |
| ➢ Diagnostic, therapeutic and theranostic | ➢ Low sensitivity |
| ➢ Increased vascular permeability | ➢ Limited targeted ability |
| ➢ Drug delivery related | ➢ Expensive |
| • ↓systemic effects | ➢ Difficult to assess the blood distribution |
| • ↓Drug-drug interaction | ➢ Discontinuation of therapy is not possible |
| • ↓Off target effects | ➢ Autonomic imbalance |
| • ↓Drug resistance | |
| • ↓drug degradation | |
| • ↑Drug bioavailability | |
| • ↑Biological efficacy | |
| • ↑Safety and effectiveness | |
| • ↑Water solubility | |
| • ↑Drug stability | |
| • ↑Drug uptake | |
| • ↑Therapeutic index | |

**Author Contributions:** S.A. and V.R.: wrote the original manuscript, H.S.: made the figures and did revision, S.K.N.: did the revision work and addressed the reviewers comments, V.R.: edited and finalized the manuscript. All authors have read and agreed to the published version of the manuscript

**Funding:** This research received no external funding.

**Institutional Review Board Statement :** Not Applicable.

**Informed Consent Statement :** Not Applicable.

**Data Availability Statement:** Not Applicable.

**Conflicts of Interest:** The authors declare no conflict of interest.

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
