# Peer review of "Nanomaterial-Mediated Theranostics for Vascular Diseases"

_jnt, doi:10.3390/jnt2010001_

Round 1

Reviewer 1 Report

The authors reviewed the possible potential of nanoparticles in atherosclerosis treatment and diagnosis.  The authors extensively reviewed therapeutic potentials in animal studies and possible target molecules based upon basic research for the recent 20 years publication.  However, last year in Alyssa M. Flores et al reviewed this topic in Arteriosclerosis, Thrombosis, and Vascular Biology. 2019;39:635–646 and could not find new findings since then to add.  In contrast, the view of nanoparticles in diagnostic tools is rather new and several updated articles are published.  I recommend the authors to focus in point of view in diagnostic tools.  A recent review of nanoparticle and atherosclerosis diagnosis is focused on CAD but not PAD (Cardiovasc Drugs Ther. 2016 Feb; 30(1): 33–39, Theranostics. 2018; 8(17): 4710–4732).  It will make this review novel and attractive to the readers.

Author Response

Comment: The authors reviewed the possible potential of nanoparticles in atherosclerosis treatment and diagnosis.  The authors extensively reviewed therapeutic potentials in animal studies and possible target molecules based upon basic research for the recent 20 years publication.  However, last year in Alyssa M. Flores et al reviewed this topic in Arteriosclerosis, Thrombosis, and Vascular Biology. 2019;39:635–646 and could not find new findings since then to add.  In contrast, the view of nanoparticles in diagnostic tools is rather new and several updated articles are published.  I recommend the authors to focus in point of view in diagnostic tools.  A recent review of nanoparticle and atherosclerosis diagnosis is focused on CAD but not PAD (Cardiovasc Drugs Ther. 2016 Feb; 30(1): 33–39, Theranostics. 2018; 8(17): 4710–4732).  It will make this review novel and attractive to the readers.

Response: Thank you for your comments and suggestions. We have added the text on the diagnostics in relation to PAD (lines 248-290) and have added Table 2. We have also included the ongoing clinical trials related to nanoparticle and PAD (Lines 322-332).

Reviewer 2 Report

The authors wrote a review paper that covers briefly the application of nanomaterials, including nanoparticles, for treatment of atherosclerosis and other vascular diseases. I have some suggestions for a major revision:

  1. In Fig.1 the letters inside the figure are blurred and not readable.
  2. Fig. 2 looks very similar to Fig.1. To avoid repetitions I could suggest to merge the Fig.1 and 2 and to make panels (a) and (b) for the differences.
  3. The authors could include a new figure dedicated to the topic of the article "Diagnostic role in vascular diseases". At the moment this section looks unfinished.
  4. The section “Advantages and limitations” could be appended with a table listing the advantages and the limitations for a straightforward presentation to the reader.
  5. The list of references should be appended with a recent review on the topic: Dual and multi-drug delivery nanoparticles towards neuronal survival and synaptic repair. Neural Regen Res 2017 ;12:886-9. There are many common nanomaterials for disease treatment between the field of atherosclerosis and the synaptic repair.
  6. The article does not have an acknowledgement to any funding agency. Is it the case or authors forgot to include one?

Author Response

Comment: The authors wrote a review paper that covers briefly the application of nanomaterials, including nanoparticles, for treatment of atherosclerosis and other vascular diseases.

Response: Thank you for your comment.

I have some suggestions for a major revision:

Concern 1: In Fig.1 the letters inside the figure are blurred and not readable.

Response: Thank you for your comment. We have modified the Figure 1 in the revised manuscript.

Concern 2: Fig. 2 looks very similar to Fig.1. To avoid repetitions I could suggest to merge the Fig.1 and 2 and to make panels (a) and (b) for the differences.

Response: Thank you for your comment and suggestion. We have merged Figures 1 and 2 in the revised manuscript.

Concern 3: The authors could include a new figure dedicated to the topic of the article "Diagnostic role in vascular diseases". At the moment this section looks unfinished.

Response: Thank you for your comment and suggestion. We have included a new figure (Figure 2) for the Diagnostic section in the revised manuscript as suggested by the reviewers. We have also modified the text for this section and have included more studies on PAD.

Concern 4: The section “Advantages and limitations” could be appended with a table listing the advantages and the limitations for a straightforward presentation to the reader.

Response: Thank you for your comment and suggestion. We have included Table 3 in the revised manuscript.

Concern 5: The list of references should be appended with a recent review on the topic: Dual and multi-drug delivery nanoparticles towards neuronal survival and synaptic repair. Neural Regen Res 2017 ;12:886-9. There are many common nanomaterials for disease treatment between the field of atherosclerosis and the synaptic repair.

Response: Thank you for your comment and suggestion. We have included the article in the revised manuscript and appended the review in the citation list.

Concern 6: The article does not have an acknowledgement to any funding agency. Is it the case or authors forgot to include one?

Response: Thank you for your comment. This project was not funded by any agency.

Round 2

Reviewer 1 Report

The authors edited properly and now the article can be accepted as it is.

Reviewer 2 Report

The authors did the changes that I requested. i do not have other remarks.